# Upper Respiratory Microbiome in Pregnant Women: Characterization and Influence of Parity

**DOI:** 10.3390/microorganisms10112189

**Published:** 2022-11-04

**Authors:** Giulia Solazzo, Simona Iodice, Jacopo Mariani, Nicola Persico, Valentina Bollati, Luca Ferrari

**Affiliations:** 1EPIGET LAB, Department of Clinical Sciences and Community Health, Università degli Studi di Milano, 20122 Milan, Italy; 2Department of Clinical Sciences and Community Health, Università degli Studi di Milano, 20122 Milan, Italy; 3Department of Obstetrics and Gynecology, Fondazione IRCCS Ca’ Granda Ospedale Maggiore Policlinico, 20122 Milan, Italy; 4Occupational Health Unit, Fondazione IRCCS Ca’ Granda Ospedale Maggiore Policlinico, 20122 Milan, Italy

**Keywords:** respiratory microbiome, pregnancy, parity

## Abstract

During pregnancy, the woman’s immune system changes to support fetal development. These immunological modifications can increase the risk of respiratory diseases. Because the respiratory microbiome is involved in airway homeostasis, it is important to investigate how it changes during pregnancy. Additionally, since parity is associated with immune system alterations and cohabitants shared a similar microbiome, we investigated whether having a child may influence the respiratory microbiome of pregnant women. We compared the microbiome of 55 pregnant with 26 non-pregnant women using 16S rRNA gene sequencing and analyzed taxonomy, diversity, and metabolic pathways to evaluate the differences among nulliparous, primiparous, and multiparous women. The microbiome was similar in pregnant and non-pregnant women, but pregnant women had higher alpha diversity (Chao1 *p*-value = 0.001; Fisher *p*-value = 0.005) and a lower abundance of several metabolic pathways. Multiparous pregnant women had a higher relative abundance of *Moraxella* (*p*-value = 0.003) and a lower abundance of *Corynebacterium* (*p*-value = 0.002) compared with primiparous women. Both multiparous (pregnant) and primiparous/multiparous (non-pregnant) women reported a higher abundance of *Moraxella* compared with primiparous (pregnant) or nulliparous ones (*p*-value = 0.001). In conclusion, we characterized for the first time the upper airway microbiome of pregnant women and observed the influence of parity on its composition.

## 1. Introduction

During pregnancy, the woman’s body significantly changes its metabolism, hormonal status, and immunological defenses to provide a healthy environment for fetal development [1]. The change in the woman’s immune system is crucial to support fetal growth and follows precise timing [2]. The modifications in the immune system, such as altered T lymphocyte immunity, increase the risk of respiratory diseases, such as acute respiratory distress syndrome (ARDS) and pneumonia, which are caused by several microorganisms (i.e., bacteria and viruses) [3]. In this scenario, the microbiome may play an important role, either protective or detrimental.

The microbiome is strictly associated with the immune system and inflammation [4] and it is well-known that it changes during pregnancy in different body sites [1]. In pregnant women, the gut and vaginal microbiomes are the most widely investigated. The first is mainly composed of *Proteobacteria* and *Actinobacteria,* while the vaginal microbiome is enriched in *Lactobacilli* during pregnancy. Both gut and vaginal microbiome dysbiosis during pregnancy have been associated with several adverse outcomes, such as pre-eclampsia and gestational diabetes [5,6,7,8].

In respiratory diseases, the microbiome of the upper airway, in synergy with the immune system, might be considered the primary line of defense against pathogens [9,10,11]; nonetheless, the composition of the respiratory microbiome in pregnant women has never been investigated so far.

The different body-site microbiomes and among them the airway microbiome are shaped by several environmental factors including the people who live with us. Several studies suggested that co-habitants share a similar microbiome composition [12,13]. A recent study described how the development of the respiratory microbiome can be shaped by having siblings [14]. In this scenario, parity might influence the respiratory microbiome of pregnant women parity. Indeed, parity has been associated with immune system modulation and inflammation [15]; it is important for fetal development and woman’s health [16,17,18,19]. and recently has been associated with the human vaginal microbiome composition [20].

In the present explorative study, we sought to characterize the composition of the upper respiratory bacterial microbiome in a population of healthy pregnant women and to investigate whether its composition might be influenced by parity.

## 2. Materials and Methods

### 2.1. Study Population

The present study involved healthy volunteer pregnant women, and non-pregnant women matched for age and body mass index (BMI). All the 81 women included in the study were recruited between 2018 and 2019 at the “Fondazione IRCCS Ca’ Granda—Ospedale Maggiore Policlinico in Milan, Italy. All participants provided signed written informed consent. The study design, research aims, and measurements were approved by the Ethics Committee “Comitato Etico—Milano Area 2” of the Fondazione IRCCS Ca’ Granda Ospedale Maggiore Policlinico, 20122 Milan, Italy (approval number #318), in agreement with principles of the Helsinki Declaration. Eligibility criteria included women aged ≥18 years, with no known chronic or infectious disease. All the pregnant women were enrolled in the first trimester [21]. Each enrolled subject filled out a questionnaire to obtain information such as age, height, weight, and smoking habits. The inclusion criteria for participants were the following: being older than 18 years at enrolment; physiological pregnancy (for pregnant women); being resident in Lombardy at the time of recruitment. Exclusion criteria included a previous diagnosis of cancer and cardiovascular diseases or other chronic diseases, such as multiple sclerosis, epilepsy, schizophrenia, Alzheimer’s disease, Parkinson’s disease, depression, and bipolar disorder.

### 2.2. Sample Collection and Sequencing

For each woman, we collected and stored at −80 °C a nasopharynx swab following standard guidelines. QIAamp UCP Pathogen Mini (Qiagen, Hilden, Germany) protocol was applied for DNA extraction. Then, we shipped the DNA samples to the sequencing service facility Personal Genomics Srl (Verona, Italy). The 16S sequencing was performed using the primers Pro341F and Pro805R to amplify the region V3-V4. The libraries were evaluated by a Labchip DNA High Sensitivity kit (Perkin Elmer, Waltham, MA, USA), quantified by the Qubit dsDNA BR assay kit (Thermofisher Scientific, Waltham, MA, USA), and finally sequenced through the Illumina MiSeq platform (Illumina, San Diego, CA, USA) using a paired-end library of 300 bp insert size.

### 2.3. Bioinformatics and Statistics

The quality of the reads was checked using FastQC v0.11.2. We analyzed the reads using QIIME 2 v2022.2 and denoised them using DADA2 pipeline. Taxonomy was assigned using a pre-trained Naïve Bayes classifier (Silva database, release 132, 99%). To estimate the diversity, the R package “phyloseq” was used, R software v3.6.2. Alpha diversity was calculated using Chao1, Shannon, and Fisher index. The difference in alpha diversity among groups was tested using Wilcoxon rank-sum test (*p*-value adjusted for Benjamini-Hochberg). The beta diversity was calculated using Bray–Curtis, Jaccard, and weighted UniFrac distances. The groups were compared through permutational multivariate analysis of variance (PERMANOVA, Adonis from the R package “vegan”) with 9999 permutations. Finally, we identified the core microbiome at 5% abundance using the R package “OTUtable”. The relative abundance of each genus among groups was tested with the Kruskal–Wallis test (adjusted for Benjamini–Hochberg FDR < 0.10) using STAMP software v2.1.3. The pathway prediction analysis was performed using the plugin q2-picrust2 in QIIME 2, then we selected the pathway detected in at least 15% of our samples. We compared the pathway abundance between pregnant and non-pregnant women with Welch’s t-test (adjusted for Benjamini–Hochberg FDR < 0.10) using STAMP software. We did the same analysis to compare the pathway abundance between multiparous and primiparous women. All the figures were created in R using the package “ggplot2” and STAMP software.

## 3. Results

The present study includes 55 pregnant women and 26 non-pregnant women. The main characteristics of the study population are reported in Table 1. None of the pregnant women included in the study had pregnancy interruptions.

### 3.1. Differences in the Nasal Microbiome between Pregnant and Non-Pregnant Women

The group of pregnant women showed a higher alpha diversity compared to the group of non-pregnant women of all three alpha diversity indices estimated: Chao1, Shannon, and Fisher (Table 2; Figure 1). Two out of three alpha diversity indices also showed a significant difference between the two groups when compared using the Wilcoxon test (Chao1 *p*-value = 0.001; Fisher *p*-value = 0.005). Likewise, we observed a significant difference in beta diversity between the two groups (PERMANOVA, Bray–Curtis *p*-value = 0.01; Jaccard *p*-value = 0.02; UniFrac *p*-value = 0.01).

Overall, the microbiome composition was similar between the pregnant and non-pregnant groups. We estimated the core microbiome at a 5% abundance threshold for each group. The relative abundance of seven genera was different between the two groups (Figure 2). *Pseudomonas, Gulbenkiania*, *Enterococcus*, and *Burkholderia–Caballeronia–Paraburkholderia* were identified only in the non-pregnant group core microbiome. These taxa were excluded from the core microbiome of the pregnant women group, as they had a mean relative abundance of ≤ 5%. On the contrary, pregnant women had a higher abundance of *Acinetobacter* and *Enhydrobacter* (*p*-value = 0.019, and 0.043 respectively), and in addition, the *Psychrobacter* genus was identified only in the core microbiome of the pregnant group. We performed pathway analysis and observed differences in the abundance of 25 pathways (Figure 3). Specifically, 18 pathways were reduced in pregnant women, while 7 were increased (i.e., PWY-6728, PWY-7031, PWY-7198, THISYN-PWY, FUC-RHAMCAT-PWY, METH-ACETATE-PWY, FUCCAT-PWY).

### 3.2. Nasal Microbiome and Parity

In the pregnant women group, 24 subjects were multiparous, while 31 were primiparous. Both alpha and beta diversity did not differ between the two groups. However, when we compared the microbiome of these groups, we observed that multiparous pregnant women had a higher abundance of *Moraxella* (Kruskal–Wallis test, *p*-value = 0.003; FDR = 0.016) (Figure 4a). On the other hand, the primiparous pregnant women had a higher abundance of *Corynebacterium* (Kruskal–Wallis test, *p*-value = 0.002; FDR = 0.026) (Figure 4b). In addition to these results, we found an increase of *Moraxella* abundance in both multiparous (pregnant) and primiparous/multiparous (non-pregnant) when compared with primiparous (pregnant) or nulliparous (Kruskal–Wallis test, *p*-value = 0.001; FDR = 0.0047) (Figure 5). No difference in pathway abundance was detected between multiparous and primiparous women.

## 4. Discussion

The respiratory microbiome is crucial for human health [11]. Indeed, it modulates the immune response and contributes to counterattack respiratory infections [22,23] and its alteration has been involved in many inflammatory and respiratory diseases [24,25,26,27]. In the present study, we characterized the nasal microbiome of 55 pregnant women and 26 non-pregnant women and observed that pregnant women reported an increased alpha diversity. In line with our findings, some studies on the oral and gut microbiome have already described an increase in bacterial communities in pregnant women [28,29]. This evidence might be a consequence of the physiological changes occurring during pregnancy. Pelzer and colleagues suggested that female hormones might promote the proliferation of specific taxa. These authors reported that steroid hormones (estradiol and progesterone) stimulate the in vitro growth of *Lactobacillus* spp., *Bifidobacterium* spp., *Streptococcus* spp., and *E. coli* [30].

Considering the microbial composition, two Gram-negative genera, *Acinetobacter* and *Enhydrobacter,* showed the highest abundance in pregnant women. *Acinetobacter* is an opportunistic pathogen and due to its multidrug resistance and persistence in the environment, its infections are a huge concern. In recent years, the presence of *Acinetobacter* in the upper respiratory system has been widely associated with SARS-CoV-2 symptoms. Indeed, among multi-drug resistant bacteria, *Acinetobacter* spp. was the most commonly reported in patients with COVID-19 [31,32,33,34]. *Enhydrobacter* is a skin-associated genus and it has been detected also in indoor air [35,36]. Increased abundance of this taxon is associated with different diseases, such as blepharitis [37].

Additionally, we found changes in the metabolic pathway of the microbiome of pregnant women when compared with non-pregnant women. In the pregnant women group, we observed a decrease in several pathways involved in degradation and fermentation, such as the fermentation of pyruvate, while we observed a significant increase in vitamin biosynthesis. The increased level of opportunistic pathogens and the variation in the metabolic pathways might be a consequence of the modifications in the immune response during pregnancy. Indeed, during pregnancy, several immunological changes occur to provide fetal tolerance and these changes make pregnant women more susceptible to both bacterial and viral infections. Viral infections (such as influenza A virus, hepatitis E virus, herpes simplex virus, and measles virus) are more severe and re-activation of latent bacterial infections is more common. In this scenario, microbiome modifications during pregnancy may contribute to maintaining homeostasis in response to these immunological alterations [38].

It is known that parity influences both pregnant woman’s health and fetal development, and a recent study identified an association between vaginal microbiome and parity [20]. In our study, we observed that multiparous women had a higher abundance of *Moraxella* compared to the primiparous group, which had a higher abundance of *Corynebacterium*. In addition, also primiparous/multiparous non-pregnant women had a higher abundance of *Moraxella* when compared to nulliparous women. Because *Moraxella* is one of the most common genera in the upper respiratory microbiome of children [10], children’s respiratory microbiomes might influence the microbiome of their mothers. Indeed, several studies suggested that people who live together share a similar microbiome [14,39]. A recent study analyzed both the airway and gut microbiome in siblings and it was reported that having siblings is a critical factor that shapes the child’s developing microbiome [14]. However, also unrelated people who share the same environment have a similar microbiome. A study in the Dutch Microbiome Project (DMP) cohort analyzed the correlation between the gut microbiome and 241 host and environmental factors (i.e., breastfed, smoking habits, PM_2.5_, and NO_2_ concentration). In the gut microbiome only, 6.6% of taxa resulted in being heritable, while the variance of 48.6% of taxa could be explained by cohabitation [13].

To our best knowledge, this is the first study to characterize the upper respiratory microbiome in pregnant women and to describe the difference between multiparous and primiparous women. We also acknowledge some limitations. First, the small sample size hampers our findings with some uncertainty and precludes definitive conclusions. Moreover, the use of 16S rRNA sequencing did not allow us to deeper investigate the microbial communities. Third, the lack of information on the children’s microbiome prevented the investigation of possible correlations between mothers’ and children’s microbiomes.

## 5. Conclusions

In the present explorative study, we report for the first time the different upper airway microbial compositions and the related pathways between pregnant and non-pregnant women. Moreover, the higher relative abundance of *Moraxella* in multiparous pregnant women and primiparous/multiparous non-pregnant women supports the influence of parity on the upper airway microbiome.

## Figures and Tables

**Figure 1 microorganisms-10-02189-f001:**
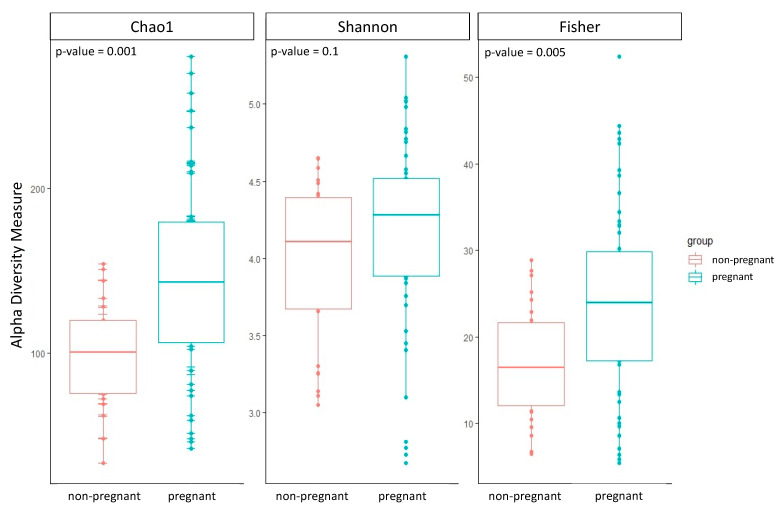
Boxplots of the three alpha diversity indices were calculated for both groups of pregnant and non-pregnant women. The *p*-value was estimated using the Wilcoxon test.

**Figure 2 microorganisms-10-02189-f002:**
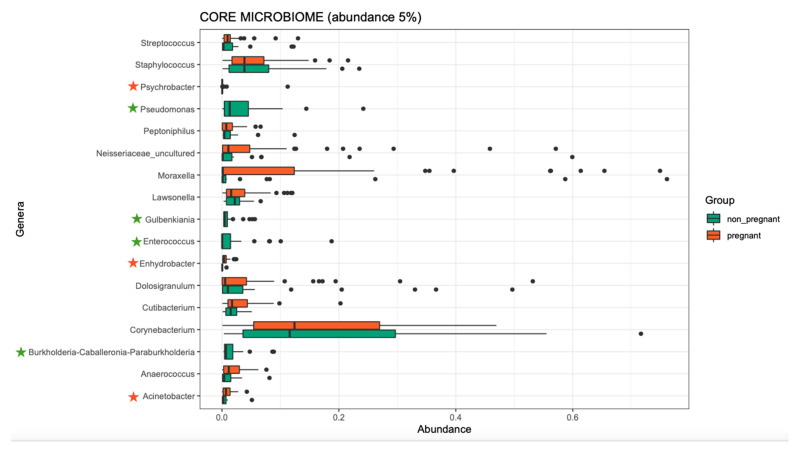
The core microbiome of pregnant women and non-pregnant women groups. Stars indicate the genera that are different between the two groups. Two genera showed higher abundance in the pregnant group, and one was identified only in the pregnant group (orange stars); four genera were identified only in the non-pregnant group (green stars).

**Figure 3 microorganisms-10-02189-f003:**
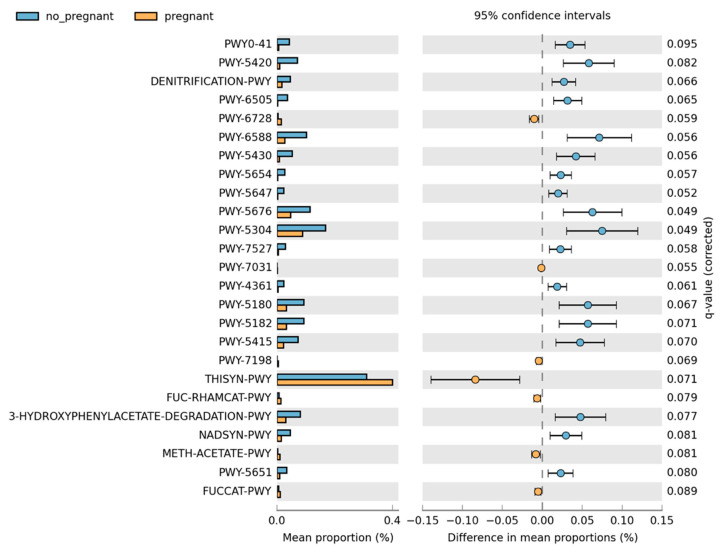
Differences in the pathway relative abundance between pregnant and non-pregnant women. The difference was estimated using Welch’s *t*-test (adjusted for Benjamini–Hochberg FDR < 0.10).

**Figure 4 microorganisms-10-02189-f004:**
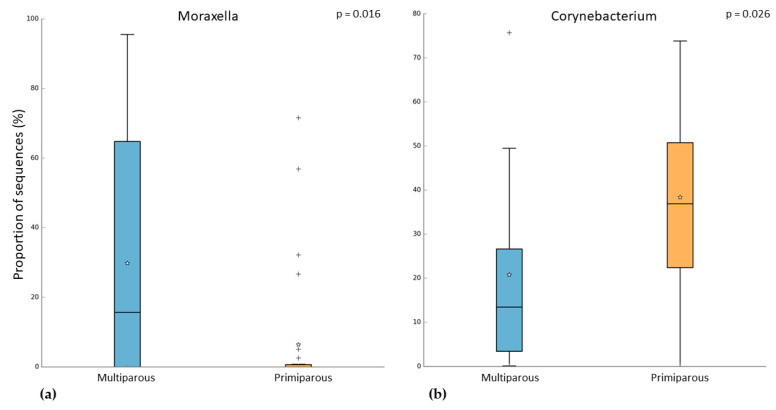
Difference between multiparous and primiparous pregnant women in the relative abundance of (**a**) *Moraxella* and (**b**) *Corynebacterium.* “+” indicate the outliers; “stars” indicate the mean value.

**Figure 5 microorganisms-10-02189-f005:**
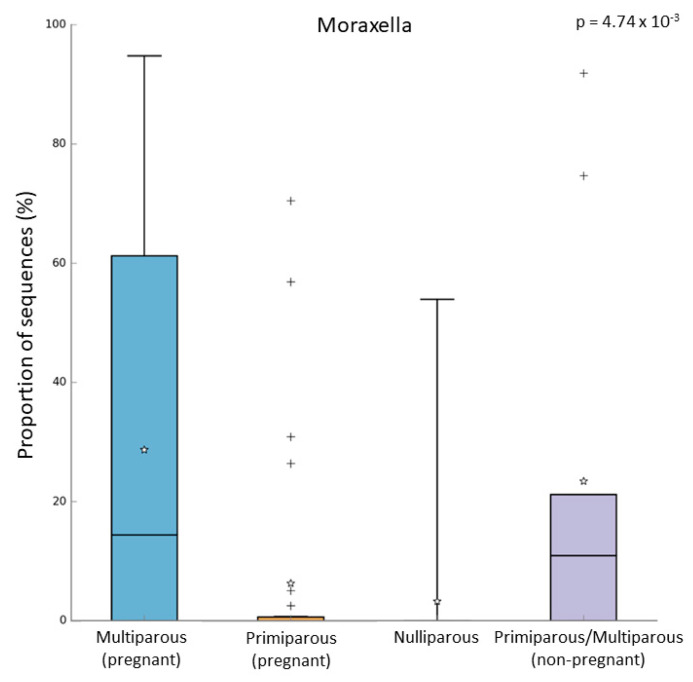
Relative abundance of *Moraxella* in multiparous (N = 24), primiparous (N = 31), nulliparous (N = 17), and non-pregnant women primiparous/multiparous (N = 9). “+” indicate the outliers; “stars” indicate the mean value.

**Table 1 microorganisms-10-02189-t001:** Characteristics of the study participants.

Characteristics	Pregnant (N = 55) Mean (±sd) or N (%)	Non-Pregnant (N = 26) Mean (±sd) or N (%)	*p*-Value
Age, years	34.2 (±3.0)	34.1 (±8.0)	0.83
BMI, Kg/m^2^	22.5 (±3.5)	21.4 (±2.8)	0.18
*<25*	44 (80%)	23 (88%)	
*≥25*	11 (20%)	3 (12%)	
Parity			0.11
*Nulliparous*	0	17 (65.4%)	
*Primiparous*	24 (44%)	2 (7.7%)	
*Multiparous*	31 (56%)	7 (26.9%)	
Smoker			0.92
*Yes*	8 (15%)	4 (20%)	
*No*	47 (85%)	16 (80%)	

**Table 2 microorganisms-10-02189-t002:** First quartile, median, and third quartile of the alpha diversity in the two groups.

Diversity Index	Pregnant (N = 55) 1st Qu. Median 3rd Qu.	Non-Pregnant (N = 26) 1st Qu. Median 3rd Qu.
Chao1	106.5 143.0 179.5	75.5 100.5 119.8
Shannon	3.88 4.28 4.52	3.67 4.11 4.40
Fisher	17.28 23.91 29.79	12.05 16.46 21.64

## Data Availability

Data will be made available on request.

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
