# Peer review of "Upper Respiratory Microbiome in Pregnant Women: Characterization and Influence of Parity"

_microorganisms, 2022, doi:10.3390/microorganisms10112189_

Round 1

Reviewer 1 Report

It is an interesting study that reports the impact of the parity on the upper airway microbiome. The authors compared the upper airway microbial compositions between pregnant and non-pregnant women. Also, the authors showed that higher relative abundance of Moraxella in multiparous pregnant women and primiparous/multiparous non-pregnant women.

Major comments

1- Please provide statistics in table #1. Also are values present represented mean (SD)?? since no SD appeared in the table

2- Figure 2: it seems that Pseudomonas is present in non -pregnant, while no bacteria in pregnant. Why the authors did not include it? 

3- Figure 4: Did the difference in microbiome affect the pregnancy outcomes? please also include this one in the table 1

Reviewer 2 Report

 This is a straightforward study, contributing to our insight into the composition of the upper respiratory microbiome of healthy pregnant women and its association with parity. The study found increased alpha diversity of the nasal microbiome in pregnant versus non-pregnant women, with the core microbiome being largely similar in pregnant and non-pregnant women. As well, a significantly higher abundance of Moraxella spp. was observed in parous versus nulliparous women, which may suggest that, since Moraxella is a commonest genus in the upper respiratory microbiome of children, women acquire this bacterium from their children. This support an important assumption made in recent studies that people living together share similar microbiomes. The study has some limitations, but these are addressed by the authors. In general, the manuscript is well-structured, methodologically sound, and clearly presented.

Author Response

We thank the reviewer 2 for her/his comments.

Round 2

Reviewer 1 Report

No further comments